# Cold-Pressed Sacha Inchi Oil: High in Omega-3 and Prevents Fat Accumulation in the Liver

**DOI:** 10.3390/ph17020220

**Published:** 2024-02-07

**Authors:** Tepparit Samrit, Supawadee Osotprasit, Athit Chaiwichien, Phawiya Suksomboon, Supanan Chansap, Anan Athipornchai, Narin Changklungmoa, Pornanan Kueakhai

**Affiliations:** 1Food Bioactive Compounds Research Unit, Faculty of Allied Health Sciences, Burapha University, Long-Hard Bangsaen Road, Saen Sook Sub-District, Mueang District, Chonburi 20131, Thailand; 62810099@go.buu.ac.th (T.S.); 63810104@go.buu.ac.th (S.O.); 63810103@go.buu.ac.th (A.C.); 61810042@go.buu.ac.th (P.S.); 62810076@go.buu.ac.th (S.C.); narinchang@go.buu.ac.th (N.C.); 2Department of Chemistry and Center for Innovation in Chemistry, Faculty of Science, Burapha University, Chonburi 20131, Thailand; anana@buu.ac.th

**Keywords:** Sacha inchi, oil supplement, fatty liver, omega-3 fatty acids

## Abstract

The ability of oil supplementation to inhibit various metabolic syndromes has been recognized. However, there are currently no studies determining the effects of oil supplements on healthy conditions. *Plukenetia volubilis* L., also known as Sacha inchi, is a seed rich in essential unsaturated fatty acids that improves metabolic syndrome diseases, such as obesity and nonalcoholic fatty liver. However, the health benefits and effects of Sacha inchi oil (SIO) supplementation remain unclear. This study aims to evaluate the chemical effects and properties of Sacha inchi oil. The results of the chemical compound analysis showed that Sacha inchi is an abundant source of ω-3 fatty acids, with a content of 44.73%, and exhibits scavenging activity of 240.53 ± 11.74 and 272.41 ± 6.95 µg Trolox/g, determined via DPPH and ABTS assays, respectively, while both olive and lard oils exhibited lower scavenging activities compared with Sacha inchi. Regarding liver histology, rats given Sacha inchi supplements showed lower TG accumulation and fat droplet distribution in the liver than those given lard supplements, with fat areas of approximately 14.19 ± 6.49% and 8.15 ± 2.40%, respectively. In conclusion, our findings suggest that Sacha inchi oil is a plant source of ω-3 fatty acids and antioxidants and does not induce fatty liver and pathology in the kidney, pancreas, and spleen. Therefore, it has the potential to be used as a dietary supplement to improve metabolic syndrome diseases.

## 1. Introduction

Nonalcoholic fatty liver (NAFLD) is a chronic liver disease worldwide. In 2020, the global prevalence of NAFLD was 25%, and it increases every year. Risk factors of NAFLD are type 2 diabetes mellitus and obesity [1]. In severe cases, NAFLD may even progress to fibrosis and liver cancer, which increases the mortality rate [2,3]. The mechanism of NAFLD development starts with simple steatosis, characterized by fat accumulation in liver tissue exceeding 5%. Fat accumulation in the liver is an important starting point for the development of NAFLD. The accumulation of triglycerides (TGs) in the liver represents the imbalance of fatty acid input and output [4]. Fatty acids are produced from nonlipid dietary, peripheral tissue (adipose), and dietary fatty acids. Intrahepatic TG can induce inflammation and oxidation in the liver, which results in steatohepatitis [5]. A recent study reported that patients with steatohepatitis have an increased risk of chronic kidney disease than patients with simple steatosis. Steatohepatitis induces glucose and lipid metabolism [6]. Pathogenic mediators, such as reactive oxygen species, proinflammatory markers, and tumor necrosis factor (TNF)-α from steatohepatitis liver, can promote kidney injury [6,7], pancreatitis [8], and splenomegaly [9].

Dietary fatty acids are essential nutrients for energy and play an important role in the development of many diseases. The qualitative or quantitative effects of fatty acids are the main factor for fatty liver [10]. High-fat and excess diets are associated with obesity and complications, such as fatty liver. Moreover, the types of fatty acids vary depending on the sources of dietary fat. Vegetable sources, such as olive oil and linseed oil, are rich in ω-6. Saturated fatty acid sources are abundant in animal oils, like lard oil and tallow, or in some vegetable oils, such as palm and coconut oil. Meanwhile, ω-3 is plentiful in fish oil and is low in other sources [10,11]. It is widely known that saturated fatty acids adversely affect health, increasing the risk of metabolic syndrome. On the other hand, the intake of unsaturated fatty acids plays a significant role in reducing fat accumulation in the liver and the incidence of NAFLD [12]. The previous study mostly focused on studying the effects of dietary lipids on a high-fat diet. However, it did not isolate one factor of metabolic syndrome, especially fatty liver. In a normal diet, humans typically consume approximately 25–30 g of fat per day, equivalent to about 4% of their diet in rodent models, which leads to changes in the accumulation of fatty acids in the blood and muscles [13]. The difference in the fat source type, equivalent to a regular diet, has been shown to impact gut health, gut microbiota, and short-chain fatty acids in middle-aged rats [14]. Moreover, it leads to alterations in the fatty acid composition in the blood, an imbalance of gut microbiota, and changes in serum lipids, which contribute to NAFLD [11,13,14,15]. In addition, oil is not only a source of fatty acids but also a source of antioxidants that dissolve in the oil, such as tocopherol. Oil plays a role in inhibiting excess reactive oxygen species in the body [16]. Previous studies have investigated the effects of Sacha inchi oil (SIO) on various aspects, including its composition and toxicity [17]. These studies provide fundamental knowledge of the properties of oil. However, the concentration and effect of these phytochemicals can vary depending on the cultivation area and extraction method, and they are generally considered safe for external use and consumption as food [18].

Moreover, SIO has pharmacological properties that make it suitable for use as a food supplement. For example, SIO has shown anticancer activity by reducing tumor mass in animal trials [19]. In addition, SIO has antidyslipidemia properties at a dose of 0.5 mL/kg, which can reduce cholesterol and triglyceride levels and increase high-density lipoprotein cholesterol (HDL-C) in Holtzman rats and BALB/c mice treated with a high-fat diet [20]. The emulsion of SIO has been found to improve the lipid metabolism in obesity-induced rats by decreasing total cholesterol (TC), TG, and low-density lipoprotein cholesterol (LDL-C) and increasing HDL-C. Animal experiments have shown that SIO can also improve gut microbiota, promote lipolysis, decrease interleukin 6 (IL-6), tumor necrosis factor-α (TNF-α), and leptin levels, and reduce oxidative stress and inflammation in liver tissue. Moreover, SIO has demonstrated antidyslipidemia properties in human subjects with dyslipidemia. Daily consumption of 15 mL of oil for four months reduced serum total cholesterol and LDL-C and increased HDL-C levels [20,21,22,23]. The oral gavage dose of SIO supplement at 2.5 mL/kg, assuming the rat’s metabolic rate is 10-fold higher than humans, is referred to as 0.25 mL/kg in humans. Therefore, a 60 kg adult human was given oil at 15 mL per day, and that dose was similar to improving dyslipidemia in the previous study [19,23]. In addition, this dose showed antioxidant, anti-inflammation, antilipidemic, and protective liver health by improving the gut microbiome in rat metabolic syndrome models induced by a high-fat diet [24,25,26]. However, the impact of the dietary lipid source derived from the SIO supplement on the fat accumulation in the liver of healthy rats has not been investigated. 

This study aimed to compare the efficacy of oil supplements on health, including SIO; extra virgin olive oil (EVOO) is high in monounsaturated fatty acids, mainly oleic acid, and lard oil (LO) is high in saturated fats. Additionally, we investigated the effects of prolonged exposure to oil supplements on body composition, plasma biochemical parameters, and histopathology in a rat model. This study will play a crucial role in guiding the selection of oil supplements as dietary supplements for healthy individuals.

## 2. Results

### 2.1. Fatty Acid Compounds of Sacha Inchi Oil 

The chemical components from the SIO are presented in Table 1 and Figure 1. A total of five fatty acids amounting to 96.01% in the SIO were identified. Among these, 92.23% were unsaturated fatty acids and 7.73% were saturated fatty acids. The major components of the unsaturated fatty acids were linolenic acid, C18:3 (ω-3, 44.73%), and linoleic acid, C18:2 (ω-6, 35.17%). The ratios of oleic acid to linoleic acid (ω-9/ω-6) and linoleic acid to linolenic acid (ω-6/ω-3) were calculated as 0.25 and 0.78, respectively.

### 2.2. Effect of Sacha Inchi Oil on Antioxidation

The free radical scavenging activity of three types of oil, namely, SIO, EVOO, and LO, was measured by comparing Trolox, a standard substance, using the DPPH and ABTS methods (Figure 2). The results of this study expressed that SIO had the highest antioxidant activity. SIO showed a % of scavenging activity between 5.52 ± 1.13 and 83.09 ± 0.68 mg/mL and an EC50 value of 50.76 ± 2.46 mg/mL using the DPPH method, and between 14.23 ± 0.80 and 91.09 ± 3.23 mg/mL with an EC50 value of 12.38 ± 0.32 mg/mL using the ABTS method. EVOO had lower antioxidant activity, which showed a % of scavenging activity ranging from 0.53 ± 0.72 to 33.74 ± 4.86 mg/mL. The EC50 of EVOO was 319.11 ± 31.52 mg/mL using the DPPH method. The antioxidant effect of EVOO using the ABTS method was between 3.30 ± 0.184 and 90.90 ± 3.08 mg/mL and the EC50 value was 19.69 ± 0.96 mg/mL. The LO had the lowest antioxidant effect, which showed 0.06 ± 0.76 to 20.39 ± 1.73 mg/mL and an EC50 value of 543.71 ± 27.40 mg/mL using the DPPH method. The ABTS scavenging effect of LO showed between 7.98 ± 0.32 and 69.80 ± 2.49 mg/mL with an EC50 value of 26.65 ± 1.91 mg/mL (Figure 2A–D).

This study compared the scavenging activities of various oils with the antioxidant efficiency of Trolox using the DPPH and ABTS methods. The EC50 value of DPPH was 12.19 ± 0.95 μg/mL and calculated by effect at 3.35 ± 0.77 to 84.52 ± 1.75 μg/mL. The ABTS scavenging was 6.28 ± 4.71 to 92.82 ± 0.27 μg/mL, and the EC50 value was 3.372 ± 0.30 μg/mL. The results showed that SIO had the highest scavenging activities. The oils included SIO, EVOO, and LO with concentrations of 240.53 ± 11.74, 38.44 ± 3.63, and 22.46 ± 1.10 µg Trolox/g, respectively. According to the ABTS scavenging test results, the values were 272.41 ± 6.95, 171.54 ± 8.61, and 126.95 ± 8.98 µg Trolox/g, respectively (Figure 2E,F).

### 2.3. Effect of Sacha Inchi Oil on Body Composition in Rats

The body weight of all rats increased weekly. The LO group had the highest body weight. From weeks 8 to 12, the body weight of the SIO group was slightly lower than that of the other groups. However, the difference in body weight of the SIO group was not significant when compared with the other groups (Figure 3A).

The total weight gain of the rats was calculated as the difference between the final and initial body weights after receiving an oil supplement via oral gavage for 12 consecutive weeks. The results are displayed in Figure 3B. Rats treated with oil supplements did not show a significant change in total weight gain. However, the SIO supplement group had slightly lower body weight gain than the other groups. The Lee index, which annotates the body mass of rats, is shown in Table 2. The Lee index of the oil supplements was not significantly different from that of the control group.

### 2.4. Effect of Sacha Inchi Oil on Serum Biochemical Parameters of Rats

The serum biochemicals are presented in Table 2. The serum creatinine and BUN levels were not different in the control and other groups. Similarly, the hepatic enzymes include ALT and AST in the serum of each group. It was not significant when compared with other groups. Although the EVOO group showed a slight increase in serum lipids, such as the TC, TG, HDL-c, LDL-c, and LDL-c/HDL-c ratio compared with other groups, it was not statistically significant.

### 2.5. Effect of Sacha Inchi Oil on Histopathology of Liver in Rats

The histopathology of the liver was assessed though H&E and Oil Red O staining (Figure 4). The histology of the liver can be divided based on histopathological components into three zones, including zone 1, which is the area around the portal triad (hepatic artery, portal vein, and bile duct). Zone 2 is the area between the portal triad and the central vein, and zone 3 is the area around the central vein. The control group did not have vesical fat droplet changes in zone 2 and zone 3. In contrast to the control group, the lobular structure of the LO group remained unchanged, but the microvesicular and macrovesicular fat droplets were found in all zones. Moreover, the highest macrovesicular fat droplet change in the liver of the LO supplement group was found in zone 1 around the portal triad and sinusoidal dilation. On the other hand, the liver histology of the EVOO and SIO groups had the smallest change compared with the control group, and just had a microvesicular fat droplet change in zone 1. However, the fat droplet change in the liver histology was stained using Oil Red O staining, which showed slight fat droplets in the EVOO and SIO groups, but the LO group was found to have the highest number and size of fat droplets. However, the EVOO and SIO oil groups found lower fat droplets than the LO group, but higher than the control group. Next, the images stained with Oil Red O were calculated as the percentage of Oil Red O relative area using ImageJ software (National Institutes of Health, USA, version 1.54g) to compare the area of fat accumulation with the total area. The results are consistent with images stained with H&E and Oil Red O. The control group had a relative area of 6.10 ± 2.09%, while the LO group had the highest relative area at 14.19 ± 6.49%. The EVOO and SIO groups had relative areas of 6.89 ± 5.59% and 8.15 ± 2.40%, respectively. Statistical analysis revealed that the LO group had a significantly higher relative area than the control group. In contrast, the EVOO and SIO groups had slightly higher relative areas than the control group (Figure 4A,B). 

### 2.6. Liver TG Accumulation in Response to Sacha Inchi Oil Supplementation

The TG was measured using a colorimetric method. The liver TG levels in the control, SIO, and EVOO groups were similar, but significantly higher in the LO group compared with the other groups (Figure 5). The correlation between blood triglycerides (TGs) and liver TGs was stronger in the control group (r = 0.6974) compared with the other groups. In the EVOO and SI groups, the correlations were r = 0.6861 and 0.4463, respectively. The LO group exhibited the lowest correlation among all groups (r = 0.3407). The plasma TG and the liver TG ratio were calculated for the purpose of comparing lipid accumulation in the tissue and plasma. The results showed that the LO group had the lowest ratio of plasma and liver TG compared with the other groups, indicating a higher accumulation of triglycerides in the liver than in the plasma (Figure 6).

### 2.7. Safety of Sacha Inchi Oil on the Histology of Kidneys, Pancreas, and Spleen in Rats

The kidneys, pancreas, and spleen were stained with H&E and examined histologically. In our results, hypertrophy was not found in any of the organs. Additionally, no differences in cell structure and tissue morphology were observed in the multiple sections studied under light microscopy. All groups showed normal kidneys with typical architectures of renal tissue, renal tubules, glomeruli, and Bowman’s capsule. They exhibited normal kidney and blood vessels within the kidneys. The pancreases of the control group and all oil supplement groups did not show differences in the islets of Langerhans, intercalated duct, blood vessels, acinus, and acinar cells. This is consistent with the histological study results of the spleen, where no differences were observed in parenchymal tissues, including the white pulp and red pulp (Figure 7).

## 3. Discussion

Oil supplements have the potential as a supplement due to their abundance of essential fatty acids and antioxidant compounds. In our result, SIO contained a higher ω-3 content and antioxidant activities than LO and EVOO. In our study, LO contained 14 g of total fat in 15 mL, which included 4.5 g of saturated fat. It had 181.5 mg of ω-3, 2482.5 mg of ω-6, and 5124.0 mg of ω-9. EVOO also contained 14 g of total fat in 15 mL, including 2 g of saturated fat, 11 g of monounsaturated fatty acids, and 1 g of polyunsaturated fatty acids. It contained 86.8 mg of ω-3, 853 mg of ω-6, and 10,900 mg of ω-9 (according to the nutrition facts label). The plants were cultured in different geographical areas, resulting in variations in their fatty acid compounds [27,28,29,30,31,32]. The SIO from Peru was reported to contain approximately 45–48% of ω-3 and around 34–35% of ω-6 [27], which closely aligned with our results (ω-3 44.73% and ω-6 35.17%). The extraction method played a crucial role in determining the composition of fatty acids in the oil. SIO extracts obtained via subcritical fluid extraction, solvent extraction, and a single-screw press contained approximately 38.3–47.8% ω-3 [33]. The supercritical carbon dioxide extraction yielded a ω-3 fatty acid content at 50.41% and a ω-6 fatty acid content at 34.08% [34], while the SIO extract obtained using the cold-press method exhibited a higher ω-3 content of about 48% [35]. When compared with our results, the ω-3 and ω-6 content of the SIO from Chiang Rai, extracted using the cold-press method, was approximately 44.73% for ω-3 and 35.17% for ω-6. These values closely resemble those of the SIO cold extracts obtained using other extraction methods. The ω-3 content in the SIO was higher than other commonly used vegetable oils, such as safflower (0.15%), sunflower (0.16%), wheat germ (1.2%), rice bran (0.45%), rapeseed (1.2%), and olive (1.6%). The plant-based ω-3 source is essential and desirable because it prevents several diseases, such as obesity, diabetes, allergies, and neurodegenerative diseases [36,37]. 

Mammals cannot convert n-6 to n-3 polyunsaturated fatty acids (PUFAs), as they lack n-3 desaturase-converting enzymes. Therefore, it is necessary to obtain ω-3 and ω-6 fatty acids from the diet or supplements. The common source of ω-3 is oil from animals, such as fish oil. This oil contains 30–50% ω-3 fatty acids, including eicosapentaenoic acid (EPA), docosapentaenoic acid (DPA), and docosahexaenoic acid (DHA), while Sacha inchi oil (SIO) contains 44.73% ω-3 fatty acids in the form of α-linolenic acid. The ω-3 and ω-6 fatty acids present in SIO are transferred to cellular phospholipids and subsequently undergo digestion by phospholipases. The α-linolenic acid is converted to EPA acid by delta-5 and delta-6 desaturase before it can have an effect on health. EPA is transferred to mediators, including series 3 prostanoids (prostaglandins, thromboxanes, and prostacyclins), series 5 leukotrienes, resolvins, and protectins, which are synthesized by cyclooxygenases (COXs) and lipoxygenases (LOXs). These mediators play an important role in controlling immunity and blood vessel health by reducing inflammation [36]. In liver tissue, previous studies have reported that EPA affects fat accumulation and the liver lipid metabolism by activating adenosine monophosphate-activated protein kinase (AMPK) and peroxisome proliferator-activated receptor α (PPARα), leading to the activation of TG β-oxidation. Additionally, EPA inhibits sterol regulatory element-binding protein 1 (SREBP1C) and carbohydrate-responsive element-binding protein (ChREBP), thereby reducing lipogenesis. Meanwhile, the α-linolenic acid metabolism is limited by the availability of converting enzymes, such as delta-6-desaturase, and elongates enzymes [37,38]. 

However, SIO is not only a rich source of ω-3 but also contains ω-6 fatty acids; ω-6 fatty acids can be converted into arachidonic acid (AA), which stimulates proinflammatory responses by producing inflammatory mediators, such as series 2 prostanoids and series 4 leukotrienes. These play a significant role in activating inflammation in tissues and the immune system. For this reason, the ω-6/ω-3 ratio plays a significant role in controlling inflammation and anti-inflammation. If the ω-6 level exceeds the ω-3 level, it can lead to inflammation in the body and contribute to various diseases [36,37]. In the case of the SIO supplement, this proportion is close to 1:1. Similarly, if the ω-6/ω-3 ratio is greater than four, it can increase the risk of various metabolic syndromes [37,39]. Research shows that a higher intake of ω-6 compared with ω-3 can lead to steatosis and steatohepatitis, depending on the increasing ratio [39]. Additionally, ω-6 can stimulate inflammation, triglyceride synthesis, oxidation, and hormone resistance (leptin and insulin resistance) [32]. In a previous study, vegetable oils had a high ω-6/ω-3 ratio, such as sunflower oil at 63:1, corn oil at 56:1, EVOO at 11.14:1, and palm oil at 33.66:1 [37]. 

The antioxidant properties of oil play a major role in inhibiting many diseases, especially nonalcoholic steatohepatitis (NASH) [40]. In recent studies, it has been found that oils contain high levels of antioxidants. The DPPH [41,42] and ABTS [43] assays are widely used for studying the antioxidant effect of oils. According to the results of this research, the efficacy of SIO in scavenging free radicals depends on the source of cultivation and roasting temperature. In this research, SIO had a higher antioxidant capacity than oil extracted using solvent extraction. In previous studies, SIO studies conducted in Thailand’s Phitsanulok and Chiang Rai provinces showed a lower free radical scavenging capacity than our study using the DPPH assay. Their results ranged from 9.36 ± 0.09 to 13.72 ± 0.47% DPPH scavenging effect [44,45,46]. SIO had the highest antioxidant capacity when compared with LO and EVOO. Previous research found that SIO contains both γ-tocopherol and δ-tocopherol. In addition, SIO consists of other phytochemicals, such as carotenoid, polyphenol, and phytosterol, that are essential in providing a high antioxidant capacity [47,48,49]. On the other hand, EVOO consists of antioxidant molecules, including phenolic alcohols, lutein, β-carotene, lignans, secoiridoids, and tocopherols [50,51,52]. In a previous study, the DPPH and ABTS of EVOO were shown to be approximately 19.522–283.137 and 132.667–475.611 μg Trolox/g, respectively [53]. This is consistent with the results we obtained from EVOO via DPPH and ABTS, which were 22.46 ± 1.10 and 171.54 ± 8.61 µg Trolox/g, respectively [20]. The DPPH and ABTS scavenging activities of LO were 38.44 ± 3.63 and 126.95 ± 8.98 µg Trolox/g. In the previous study, the LO showed an EC50 at 98.9 ± 3.4 mg/mL, which was lower than fish (251.8 ± 6.4 mg/mL) and linseed oil (83.9 ± 2.7 mg/mL) but higher than corn oil (70.3 ± 1.5 mg/mL) [54]. Both LO and EVOO have been reported to contain tocopherol. LO extracted from pork was found to consist of a slight amount of α-tocopherol [53,54]. In EVOO, α-tocopherol was more abundant than γ-tocopherol and δ-tocopherol. The amount of tocopherol in LO depends on the source of the pork feed and the extraction method used [55,56,57,58,59]. Dietary antioxidant compounds can enhance the antioxidant effect on the body, especially in liver tissue, which is a major site of oxidative stress due to abundant metabolism. Oxidative stress in the liver plays a role in the progression of NAFLD by inducing lipids to become lipotoxic and causing damage to liver tissue [60,61]. Moreover, the antioxidative properties of the oil can inhibit the free radical generation mechanism in the fatty liver from the electron transport chain and lipid peroxidation [62,63]. In mice fed a high-fat diet, *P. huayabambana* oil reduced the production of free radicals in the blood compared with the control group. It increased the expression of proteins with a high antioxidant capacity, such as catalase (CAT) [20,64]. 

In our result, the LO, EVOO, and SIO supplements did not significantly influence body weight, total WG, the nasoanal range, and Lee index when compared with the control group. In recent studies, ω-3 fatty acids reduced weight gain in mice, thereby decreasing adipose tissue [65]. Moreover, ω-3 fatty acids decrease fat accumulation by decreasing the mRNA expression levels of acyl-CoA carboxylase 1 (ACC1), stearoyl-CoA desaturase (SCD), fatty acid synthase (FAS), and SREBP-1c [66,67]. Moreover, ω-6 polyunsaturated fatty acids reduce fat accumulation in adipose tissue by decreasing the mRNA levels of UPC1, leptin, and GLUT4 in rats [68] and increasing β-oxidation in C57BL/6J mice [69,70]. According to one study, fish oil decreased body weight gain in rats with a normal or high-fat diet [71]. However, many ω-3- and ω-6-rich oils, such as SIO and linseed oil, were not decreased in the body composition, such as body weight, weight gain, and Lee index in animal models. Similarly, low-ω-3 and ω-6 oils, such as palm, leaf lard, rapeseed, and sunflower oils, were not changed in the body composition [20,71]. 

Dietary lipids can digest and absorb into the bloodstream and transfer to many tissues, such as visceral tissue and blood vessels [72,73]. In a recent study, SIO reduced cholesterol and triglyceride levels in mice with dyslipidemia induced by a high-fat diet [20]. However, our results showed that administering 2.5 mL of SIO, EVOO, and LO in rats via oral gavage did not induce dyslipidemia (TC, LDL, and TG levels), resulting in normal serum lipid profiles. On the other hand, a previous study found that fish oil (rich in ω-3 polyunsaturated fatty acids) significantly decreased LDL levels and increased HDL levels in rats given a normal diet and 50 mg of fish oil for 2 and 6 weeks [74]. Therefore, none of the oil supplements increased AST, ALT, creatinine, and BUN. 

Polyunsaturated fatty acids can decrease fat mass. Moreover, these fatty acids can prevent obesity, dyslipidemia, nonalcoholic fatty liver, and insulin resistance [75,76]. As observed in liver histology, the rats receiving LO supplementation accumulated fat in zones 1–3 in a macro- and microvesicular pattern. SIO is rich in essential oils, particularly linolenic acid, which differs from LO, in that it contains a high proportion of saturated fatty acids and oleic acid. The levels of oleic acid were consistent with the experiment results, which showed more fat accumulation in the liver of the LO supplement than the SIO and EVOO supplements. Many experimental studies have reported that ω-3 prevents nonalcoholic fatty liver by decreasing fat accumulation and liver damage [77]. On the other hand, excess fatty acids induced by saturated fatty acids in the liver are crucial pathological conditions that drive the progression of the liver to stages 2 and 3 of nonalcoholic fatty liver disease (NAFLD) [78,79,80]. In a previous study, saturated fatty acids were found to induce fat accumulation in the liver by triggering ER stress, leading to hepatic cell dysfunction [81]. This process reduces excreted fatty acid and lipolysis by decreasing β-oxidation and upregulating lipogenic genes such as stearoyl-CoA desaturase (SCD), promoting TG formation [4]. In our results, EVOO slightly increased fat accumulation, consistent with previous studies. EVOO has been found to stimulate hepatic lipid accumulation by approximately 7.5–10% without causing lipotoxicity or inducing liver inflammation [80]. The correlation between plasma TG and liver TG is highly variable due to the transfer of TG between blood, liver, adipose, and muscle tissues [82].

In experiments related to the kidneys, a high-fat diet induced kidney inflammation by damaging small vascular tissues [83,84]. Palm oil and LO induce basal blood pressure and vascular wall inflammation. Therefore, these oils induce kidney inflammation and kidney dysfunction. In our study, the histology of the kidneys treated with LO, OVO, and SIO showed a normal kidney structure [48,85]. This finding is consistent with a previous study indicating that kidney lesions may be influenced by proinflammation and oxidative stress, especially in cases of steatohepatitis, but are lower in fatty liver states [7]. In a previous study on the pancreas, it was reported that prolonged exposure to high concentrations of long-chain FFAs leads to the inhibition of insulin biosynthesis and induces apoptosis in β-cells [81,86]. However, in the current study, none of the oil supplement groups experienced β-cell apoptosis or inflammation in the pancreas. The consumption of high levels of fish oil by rats resulted in increased red cell deformity, elevated relative liver and spleen weights, and decreased serum HDL, iron, and vitamin E concentrations. This could potentially lead to adverse effects on iron accumulation in the spleen and induce toxicity [86]. Nevertheless, the groups treated with SI, EVOO, and LO did not induce toxicity in the spleen in terms of histology.

According to the results of this study, SIO is rich in polyunsaturated fatty acids, particularly ω-3 and ω-6. It also contains antioxidant compounds, such as tocopherol. The potential impacts of long-term consumption of SIO on human health relate to its nutrient profile. SIO is beneficial for lipid metabolism in tissues; it can reduce the likelihood of excessive fat accumulation and inflammation, especially in liver tissue, which is a key factor in the development of fatty liver. Furthermore, it plays a vital role in preventing the formation of free radicals and in promoting and maintaining balance in the immune system [83,84,85].

In summary, there are crucial factors that strongly support the claim that SIO supplementation benefits health, especially as a source of ω-3 and antioxidants. Additionally, SIO did not induce fatty liver and pathology in the liver, kidney, pancreas, and spleen. In contrast, EVOO and LO exhibit lower antioxidant properties, especially LO, which significantly increases lipid droplet and TG accumulation. 

## 4. Materials and Methods

### 4.1. Oil Preparation

Sacha inchi fruits were obtained from TAI.C.M.S. Standard Industrial Co., Ltd., Mueang Chiang Rai, Thailand. Sacha inchi was cultivated in Chiang Rai, Thailand, in sandy loam soil. The fertilization period began with 400–600 g of manure fertilizer, which was supplemented with 200–400 g of manure every month. The fruit of Sacha inchi was harvested at 5–6 months. The Sacha inchi brown fruits (3 months) were peeled. Next, the Sacha inchi seeds without peel were dried at 120 °C for 10 min, and SIO was extracted using the cold press method in 2021. The EVOO is extra virgin olive oil that was purchased from the supermarket produced in Spain. The EVOO was extracted using the cold-press method in 2021. The LO (rendered lard oil) was produced in Thailand and purchased from a supermarket in 2021.

### 4.2. Fatty Acid Compound Analysis

The oil samples were prepared by dissolving 10 g of oil in 1 mL of methanol. The chemical composition of the oil sample was determined via gas chromatography coupled with mass spectrometry (GC-MS) using an Agilent 8890 gas chromatograph (Agilent Technologies, Inc., Santa Clara, CA, USA) equipped with an Agilent 5973 N inert mass selective (IMS) detector. A capillary column DBWAX (30 m length × 0.25 mm diameter with 0.5 µm film thickness) was employed for analysis. Helium (He) was used as carrier gas at a flow of 1.0 mL/min in a spitless mode, and both the IMS detector and the injector port temperatures were 320 and 240 °C, respectively. The elution program started at a temperature of 100 °C and was held for 2 min, and then increased to 240 °C at a speed of 10 °C/min and was held for 30 min. The MS ion source temperature was 230 °C. The electron energy was 70 eV and the solvent delay was 3 min. The scan mode was evaluated at 35–1000 *m*/*z*, a speed of 1.562 μ/s, and a 1.6 scans/s frequency. The results were evaluated via GC-MS, and semiquantitative analysis was performed using the peak area normalization method. Compounds were identified based on retention time, relative molecular weight, relevant literature, and other information of reference in comparison with MS spectra from the National Institute of Standards and Technology (NIST) mass spectral library. This experiment was conducted in 2021. 

### 4.3. DPPH and ABTS Scavenging Assays

The antioxidation effect of SIO, LO, and EVOO was determined via DPPH assay. The DPPH reagent (2,2-diphenyl-1-picrylhydrazyl; EMD Millipore, Darmstadt, Germany) was dissolved in absolute ethanol at 5 mM and diluted to 0.2 mM, which was used as a working solution. The samples were diluted with absolute ethanol to determine the % of DPPH scavenging activity. For the control, the 0.2 mM of DPPH was mixed with absolute ethanol at 1:1 in 96-well plates. For the sample, three types of oil were mixed with 0.2 mM of DPPH and used instead of oil. Trolox reagent (6-hydroxy-2,5,7,8-tetramethylchroman-2-carboxylic acid; Sigma-Aldrich, St. Louis, MO, USA) was used as a positive control. Absorbance was determined after 20 min at 518 nm. The scavenging activities were calculated according to: 
% of DPPH scavenging activity = (A control − A sample)/(A control) × 100.
(1)


The oil samples were analyzed for antioxidant effect via the ABTS method. ABTS reagent at a concentration of 7 mM (2,2′-azino-bis (3-ethylbenzthiazoline-6-sulphonic acid; EMD Millipore, Darmstadt, Germany) was induced to ABTS+ free radical reagent by incubating with 2.45 mM of potassium persulfate for 16 h at RT in the dark. ABTS+ was diluted to OD 0.70 ± 0.05 at 734 nm, which was used as a working solution. Next, the samples were mixed with ABTS+ working solution (sample), and absolute ethanol was mixed with ABTS+ working solution (control). For the positive control, the Trolox reagent was mixed with an ABTS+ working solution. The absorbance of ABTS radical scavenging was determined after 5 min at 734 nm. The percentage of ABTS radical scavenging was determined by this equal: % of ABTS scavenging activity = (control − sample)/(control) × 100. (2)


The percentages of the scavenging effect at each concentration were plotted against the standard curve. Linear equations were generated from the standard curve, and the half maximal effective concentration (EC50) was calculated by substituting the % of scavenging effect with 50. The concentration of oil samples was then calculated and compared with the concentration of Trolox to analyze the value of mg of the Trolox/g oil sample for comparing the oil’s scavenging effect. The antioxidation effect of SIO, LO, and EVOO was conducted in 2021.

### 4.4. Animals and Study Design

Seven-week-old male Sprague–Dawley rats weighing 250–300 g were purchased from Nomura Siam International and housed in a laboratory animal center at Thammasat University, Thailand. After one week of acclimatization, 24 rats were randomly divided into 4 groups (*n* = 6). The animals were housed in square polysulfone cages (26.6 × 42.5 × 18.5 cm^3^) at a constant room temperature of 22 ± 1 °C, and relative humidity was 30–70% with a 12 h light and 12 h dark cycle. The standard rodent chow and filtered water were provided ad libitum. All rats (4 groups; control, LO, EVOO, and SIO groups) were treated via oral gavage with 2.5 mL/kg body weight/day of distilled water (control), LO, EVOO, and SIO, respectively, for 12 consecutive weeks. Body weight, nasoanal length, and food intake were monitored every two weeks. Next, the Lee index of rats was calculated by body weight and nasoanal length. At the end of the feeding period, the rats were denied food for 16 h, and blood was collected from intracardiac puncture under thiopental sodium anesthesia. After the anesthesia, liver, kidney, pancreas, and spleen tissues were collected, isolated, and risen with phosphate-buffered saline. The tissues were weighted and dissected. All animal procedures were conducted under ethical principles and guidelines for using animals by the National Research Council of Thailand and approved by the Animal Care and Use Committee of Thammasat University (012/2021) and the Burapha University Institutional Animal Care and Use Committee (IACUC 008/2564). The animal study was conducted in 2022. 

### 4.5. Biochemical Measurement 

Blood samples were collected in a serum clot activator tube and centrifuged at 4000× *g* for 5 min. The serum was collected and maintained at −20 °C for blood biochemical analysis. The blood biochemical analysis included creatinine, blood urea nitrogen (BUN), alkaline phosphatase (ALP), alanine aminotransferase (ALT), aspartate transferase (AST), TC, TGs, HDL-C, LDL-C, and serum glucose were measured using a DRI-automate clinical chemistry analyzer in the laboratory animal center of Thammasat University, Thailand. The hepatic TG was evaluated from liver tissue. An amount of 100 mg of liver tissue of each rat was washed with cool PBS and homogenized using 5% NP-40 (Merck Millipore, Germany) [87]. Then, the mixture was slowly heated to 100 °C in a water bath for 5 min and centrifuged at 10,000× *g* for 5 min to remove the insoluble precipitate. The supernatants were diluted 10-fold with distilled water. The quantity of TG in hepatic tissue was measured using the colorimetric method with the Triglyceride Assay Kit (Ab6533 Abcam, Cambridge, MA, USA) following the manufacturer’s protocol. 

### 4.6. Histological Analysis 

The organs from each rat, including the liver, kidney, pancreas, and spleen, were fixed in 4% buffered paraformaldehyde and embedded in paraffin. The tissue samples were then dehydrated by immersion in 70% ethanol for 1 h, 80% ethanol for 1.5 h, 90% ethanol for 1.5 h, 95% ethanol for 1.5 h, 100% ethanol three times for 1 h, xylene three times for 1 h, paraffin twice for 2 h, and finally embedded in paraffin. After embedding, the tissue samples were sectioned to a thickness of 5 µm using a manual rotary microtome (Leica, Wetzlar, Germany). Next, hematoxylin and eosin (H&E) staining was applied to the sections. During this process, the slides were immersed in xylene for 5 min, 2 times; absolute ethanol for 3 min, 2 times; 95% ethanol for 3 min; 80% ethanol for 3 min; 70% ethanol for 3 min; tap water for 5 min; Harris hematoxylin for 3 min; tap water for 15 min (until the water was clear); 70% ethanol for 3 min; 80% ethanol for 3 min; 95% ethanol for 3 min; alcoholic eosin for 2 min; 95% ethanol for 2 min, 2 times; absolute ethanol for 3 min, 2 times; xylene for 3 min, 2 times. Finally, the slides were mounted with Bio Mount™ (Bio-Optica, Milan, Italy) and images were captured (Olympus BX43 and DP23-AOU, Tokyo, Japan). The morphology and pathology of tissues were evaluated using a light microscope. Six fields from each section were randomly selected for assessment, and two sections per group were analyzed [24]. The pathology was analyzed and evaluated by Tepparit Samrit. The pathology was graded based on the nonalcoholic fatty liver disease activity and tubular injury scores. The nonalcoholic steatohepatitis Clinical Research Network system for scoring activity and fibrosis in nonalcoholic fatty liver disease was graded on a scale from 0 to 3 (steatosis grade, 0: <5%, 1: 5–33%, 2: 34–66%, and 3: >66%; lobular inflammation, 0: none, 1: <2, 2: 2–4, 3: >4; hepatocyte ballooning, 0: none, 1: few ballooned cells, and 2: many ballooned cells) [88]. Tubular injury score (swelling of the tubular cells, loss of the brush boundary, or nuclear condensation) was graded on a scale from 0 to 4 (0, no change; 1, changes affecting 25% of the section; 2, changes affecting 25% to 50%; 3, changes affecting 50% to 75%; 4, changes affecting 75% to 100%) [89]. 

The liver samples, obtained from frozen tissue, were utilized to determine fat accumulation using Oil Red O staining. The frozen liver tissue was sectioned using cryosection. The frozen sections were air-dried on glass slides for 30 min and then fixed in 4% paraformaldehyde for 5 min. Subsequently, the slides were immersed in 60% isopropanol for 1 min before being stained with 1.5 mg/mL of Oil Red O (O0625, Sigma Aldrich) for 10 min. After staining, the slides were once again dipped in 60% isopropanol for 1 min. Finally, the slides were washed with deionized water for 10 dips and then mounted using 80% glycerol. The fat accumulation in the liver was evaluated using an ImageJ analyzer. This experiment was conducted in 2022. 

### 4.7. Statistical Analysis

The results were expressed as mean ± standard error of the mean (SEM). Comparisons between groups were performed using a one-way ANOVA analysis. Differences between individual treatment groups were compared using Tukey’s test. Statistical significance was set at *p* < 0.05. The results and charts were analyzed and drawn using GraphPad Prism software (Graphism, Inc., USA, version 7). The correlation between liver and plasma TG was analyzed for each group using Pearson correlation. The correlation coefficients were assessed using linear extrapolation of tabulated data.

## 5. Conclusions

The results confirm that the SIO, harvested and cultivated in Thailand, contains beneficial fatty acid compounds such as ω-3 and ω-6 and potential antioxidant activity. This study found that oil supplementation had no effect on body weight, total weight gain, and Lee index when compared with the control groups. Additionally, oral gavage of the oil supplement did not increase serum lipid, liver, and kidney enzyme levels in the blood plasma. Regarding histology, SIO and EVOO did not increase fat droplets and TG accumulation, while LO significantly increased fat droplets and TG accumulation in the liver. However, all oil supplements did not induce lesions and inflammation in the kidney, pancreas, and spleen. This study confirms that SIO from Thailand is rich in beneficial fatty acids, like ω-3 and ω-6, and exhibits antioxidant potential, but it does not impact body weight, fat accumulation, or cause liver inflammation. 

## Figures and Tables

**Figure 1 pharmaceuticals-17-00220-f001:**
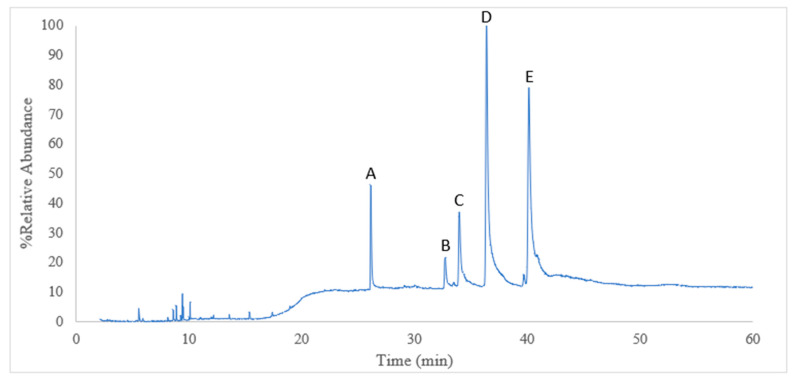
GC-MS chromatogram of fatty acids in lipid samples of SIO. Peaks of fatty acids were separated by time point. The peaks and peak areas were compared to mass spectra with those listed in the NIST2017 libraries and with published data (%). The chemical compounds of the SIO were as follows: 5.02% palmitic acid (A), 2.44% stearic acid (B), 8.65% oleic acid (ω-9) (C), 35.17% linoleic acid (ω-6) (D), and 44.73% linolenic acid (ω-3) (E).

**Figure 2 pharmaceuticals-17-00220-f002:**
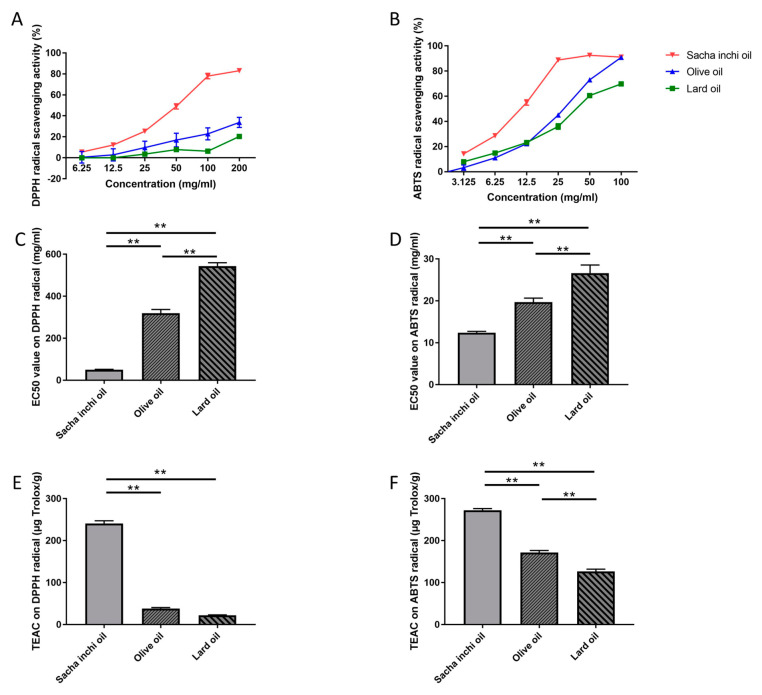
Antioxidant effects of oil supplements. The scavenging activity of SIO, EVOO, and LO on DPPH (**A**) and ABTS (**B**) is illustrated. The EC50 values for DPPH (**C**) and ABTS (**D**), along with the TEAC on DPPH (**E**) and ABTS (**F**), are provided. The results are represented with ±SEM (TEAC, Trolox equivalent antioxidant capacity; EC50, half-maximal effective concentration). ** Significance is denoted at *p* < 0.01.

**Figure 3 pharmaceuticals-17-00220-f003:**
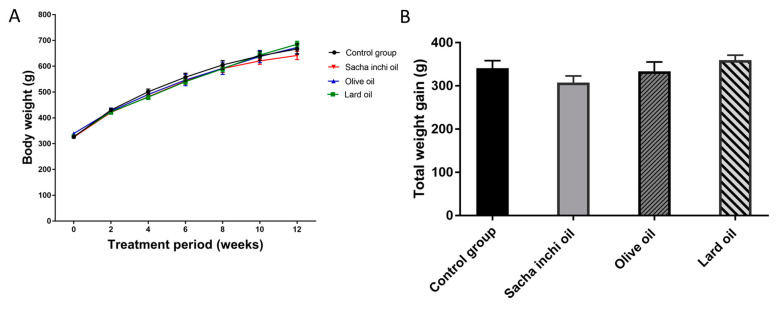
The evolution of rat body weight during the 12 weeks (**A**) and total weight gain (**B**) of rats supplemented with the control group, LO, EVOO, and SIO (±SEM) did not show statistical significance at *p* < 0.05 when compared with the control group, as analyzed by one-way ANOVA.

**Figure 4 pharmaceuticals-17-00220-f004:**
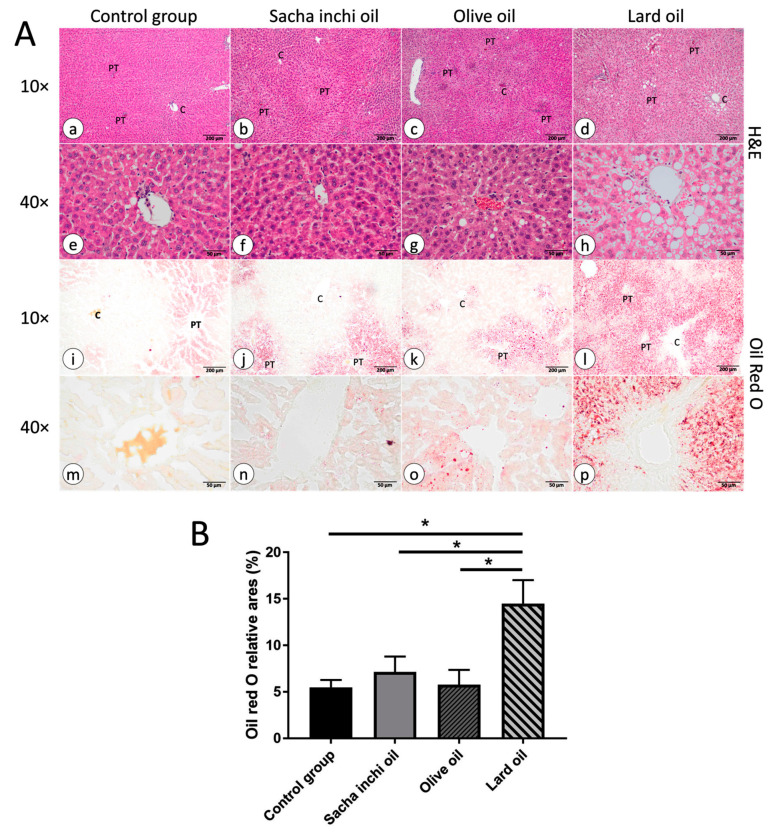
Histopathological analysis of the liver using H&E and Oil Red O Staining at 10× and 40× magnification (**A**). The control group *(n* = 6) displayed a normal liver structure (**a** and **i**, 10×; **e** and **m**, 40×). The SIO (*n* = 5; **b** and **j**, 10×; **f** and **n**, 40×) and EVOO (*n* = 6; **c** and **k**, 10×; **g** and **o**, 40×) groups exhibited slight fat droplet accumulation. The LO group (*n* = 5; **d** and **l**, 10×; **h** and **p**, 40×) showed diffuse microvesicular and macrovesicular fat droplets in the liver tissue (PT, portal triad; C, central vein). The effect of different oil types on fatty accumulation in the liver was quantified as the percentage of Oil Red O relative areas from Oil Red O-stained sections (**B**). * *p* < 0.05 indicates statistical significance when compared with the other groups.

**Figure 5 pharmaceuticals-17-00220-f005:**
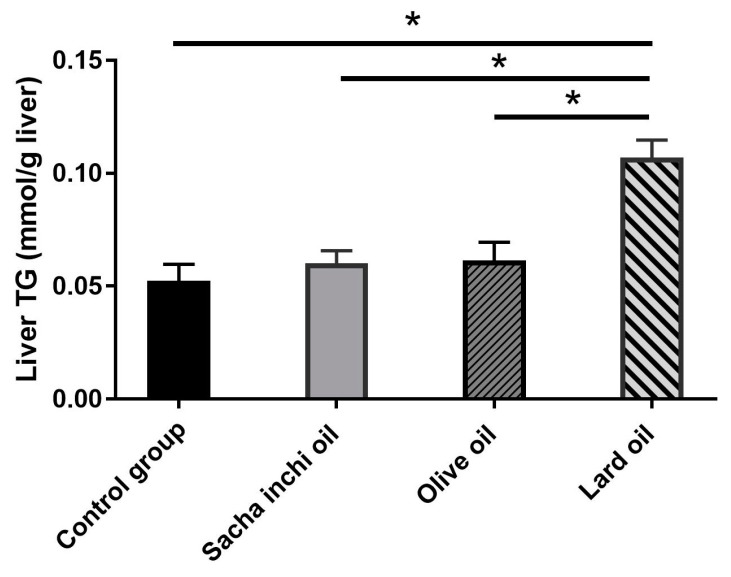
Analysis of liver triglycerides (TGs) by comparison with a TG standard. The four groups, including the control group (*n* = 6), SIO (*n* = 5), EVOO (*n* = 6), and LO (*n* = 5), are presented as mean ± SEM. * Statistical significance at *p* < 0.05 is shown when compared with the other groups, as analyzed by one-way ANOVA.

**Figure 6 pharmaceuticals-17-00220-f006:**
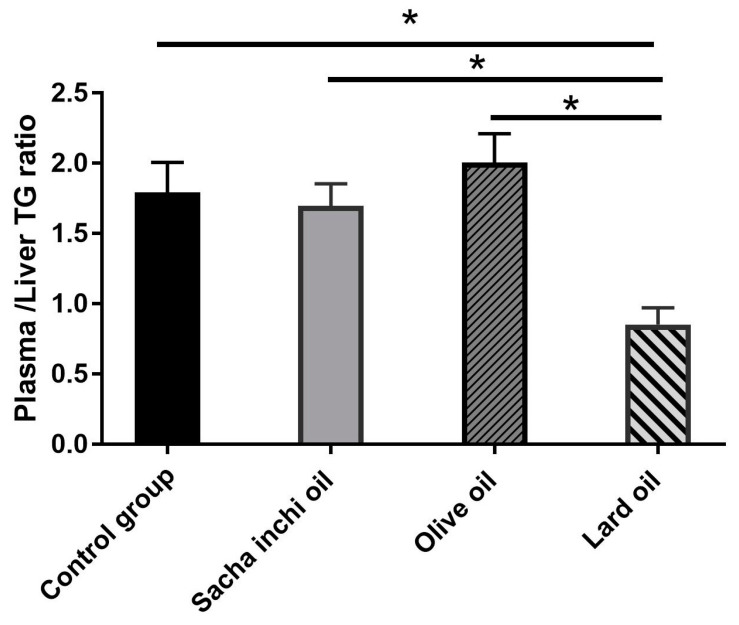
The differential plasma and liver TG values in rats were treated with an oil supplement. The results are presented as mean ± SEM. * Statistical significance at *p* ≤ 0.05 is shown when compared with the other groups, as analyzed by one-way ANOVA.

**Figure 7 pharmaceuticals-17-00220-f007:**
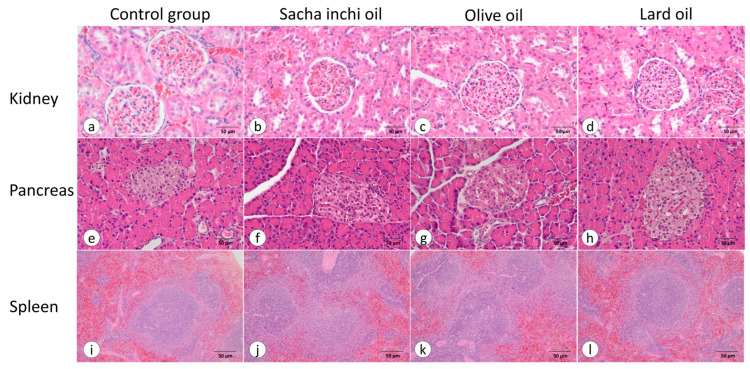
Histopathology analysis of the kidney, pancreas, and spleen using H&E staining. The control group (*n* = 6) is represented at 40× magnification in kidney (**a**), pancreas (**e**), and spleen (**i**). The SIO group (*n* = 5) is displayed at 40× magnification in the kidney (**b**), pancreas (**f**), and spleen (**j**). The EVOO group (*n* = 6) is illustrated at 40× magnification in the kidney (**c**), pancreas (**g**), and spleen (**k**). The LO group (*n* = 5) is depicted at 40× magnification in the kidney (**d**), pancreas (**h**), and spleen (**l**).

**Table 1 pharmaceuticals-17-00220-t001:** The fatty acid compounds of Sacha inchi oil analysis by the GC-MS method.

Peak No.	Retention Time (min)	Compounds		Peak Area (%) *
A	26.133	Palmitic acid	C16:0	5.02
B	32.724	Stearic acid	C18:0	2.44
C	33.952	Oleic acid	C18:1 (ω-9)	8.65
D	36.388	Linoleic acid	C18:2 (ω-6)	35.17
E	40.137	Linolenic acid	C18:3 (ω-3)	44.73
	Identified Components (%)			96.01
	Unidentified Components (%)			3.99
	Total Saturated Fatty Acid (%)			7.77
	Total Unsaturated Fatty Acid (%)			92.23
	Monounsaturated fatty acids (%)			9.01
	Polyunsaturated fatty acids (%)			83.22
	Oleic: Linoleic acid (O/L) ratio (ω-9/ω-6)			0.25
	Linoleic: Linolenic acid (L/Ln) ratio (ω-6/ω-3)			0.78

* Comparison of mass spectra with those listed in the NIST2017 libraries and with published data.

**Table 2 pharmaceuticals-17-00220-t002:** Effects of normal diet, LO, EVOO, and SIO consumption on body composition and serum biochemical parameters of rats.

Variable	Control Group	Sacha Inchi Oil	Olive Oil	Lard Oil
General physical				
Initial body weight (g)	326.17 ± 7.28	325.00 ± 7.48	329.17 ± 9.43	327.83 ± 9.28
Final body weight (g)	666.67 ± 49.72	641.80 ± 38.93	672.50 ± 58.97	685.00 ± 29.72
Total weight gain (g)	340.50 ± 43.41	307.5 ± 34.41	333.33 ± 53.09	359.2 ± 26.31
Nasoanal length (cm)	29.33 ± 0.61	28.60 ± 0.65	28.83 ± 0.98	29.00 ± 0.35
Lee index	297.71 ± 7.58	301.63 ± 9.29	303.70 ± 2.92	303.95 ± 5.03
Serum biochemical parameters				
Creatinine (mg/dL)	0.35 ± 0.04	0.34 ± 0.04	0.34 ± 0.04	0.37 ± 0.04
BUN (mg/dL)	16.30 ± 2.33	16.48 ± 1.16	16.57 ± 1.83	13.26 ± 2.38
ALT (U/L)	24.50 ± 3.62	23.20 ± 3.83	25.80 ± 2.59	20.60 ± 2.30
AST (U/L)	127.17 ± 37.36	93.00 ± 26.83	115.60 ± 21.84	106.80 ± 26.77
TC (mg/dL)	51.50 ± 7.34	50.20 ± 2.59	62.83 ± 10.93	52.20 ± 7.56
TG (mg/dL)	90.83 ± 30.20	81.40 ± 21.22	135.67 ± 58.40	79.80 ± 25.53
HDL-c (mg/dL)	28.83 ± 4.79	29.80 ± 1.92	32.50 ± 7.58	29.00 ± 5.83
LDL-c (mg/dL)	11.33 ± 2.07	11.40 ± 1.52	13.33 ± 3.78	10.80 ± 2.49
LDL-c/HDL-c ratio	0.40 ± 0.07	0.34 ± 0.04	0.41 ± 0.04	0.38 ± 0.07

## Data Availability

Data are contained within the article.

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
