# Peer review of "Cold-Pressed Sacha Inchi Oil: High in Omega-3 and Prevents Fat Accumulation in the Liver"

_pharmaceuticals, 2024, doi:10.3390/ph17020220_

Round 1

Reviewer 1 Report

Comments and Suggestions for Authors

The paper investigates the effects of Sacha Inchi oil (SIO) supplementation on health, particularly focusing on its ability to inhibit various metabolic syndromes. The study presents a detailed analysis of SIO's chemical composition, revealing a high content of ω-3 fatty acids and antioxidants. Experiments conducted on rats demonstrated that SIO supplementation does not significantly affect body weight or serum biochemical parameters. Furthermore, it was found that SIO does not induce fatty liver, nor does it affect the health of the kidney, pancreas, and spleen. These findings suggest that SIO, with its rich ω-3 and antioxidant profile, could potentially be used as a dietary supplement for improving metabolic health without causing detrimental side effects. Comments are shown below:

1. What are the potential impacts of long-term consumption of SIO on human health?

2. How does SIO compare with fish oil, a common source of ω-3 fatty acids, in terms of health benefits and bioavailability?

3. The paper does not delve deeply into the biochemical or molecular mechanisms through which SIO exerts its effects.

4. Please clarify the sample preparation process for GC-MS.

5. Did the authors compare different extraction or analysis methods?

6. The formatting of citations and references appears inconsistent. “[37-33]”

7. Please define all the abbreviations.

Author Response

Dear Professor

Thank you very much for your kind consideration, comments, and suggestions. All the suggested corrections / additions in the revised manuscript have been made and colored blue. Followings are answers to the reviewers’ comments:

Comments and Suggestions for Authors

The paper investigates the effects of Sacha Inchi oil (SIO) supplementation on health, particularly focusing on its ability to inhibit various metabolic syndromes. The study presents a detailed analysis of SIO's chemical composition, revealing a high content of ω-3 fatty acids and antioxidants. Experiments conducted on rats demonstrated that SIO supplementation does not significantly affect body weight or serum biochemical parameters. Furthermore, it was found that SIO does not induce fatty liver, nor does it affect the health of the kidney, pancreas, and spleen. These findings suggest that SIO, with its rich ω-3 and antioxidant profile, could potentially be used as a dietary supplement for improving metabolic health without causing detrimental side effects. Comments are shown below:

Comment 1 What are the potential impacts of long-term consumption of SIO on human health?

Response: Thank you sincerely for your valuable feedback. We have added the potential impacts of long-term consumption of SIO following your suggestion (line 401-408).

Comment 2 How does SIO compare with fish oil, a common source of ω-3 fatty acids, in terms of health benefits and bioavailability?

Response: We have thoughtfully and diligently made the adjustments as advised (line 276-281).

Comment 3 The paper does not delve deeply into the biochemical or molecular mechanisms through which SIO exerts its effects.

Response: We added the molecular mechanisms in SIO such as ω-3 and ω-6 on mechanism in discussion (line 281-292 and 294-298).

Comment 4 Please clarify the sample preparation process for GC-MS.

Response: We added the sample preparation process for GC-MS (line 427).

Comment 5 Did the authors compare different extraction or analysis methods?

Response: Yes, I have incorporated a section in the discussion that compares the fatty acid compounds obtained through different extraction methods (line 261-269).

Comment 6 The formatting of citations and references appears inconsistent. “[37-33]”

Response: We have made the adjustments based on your suggestions (line 259).

Comment 7 Please define all the abbreviations.

Response: We are immensely grateful for your advice that has helped improve our work, and we have already added the content as you suggested (line 77-80, and 145-147).

Sincerely yours,

Tepparit Samrit

Reviewer 2 Report

Comments and Suggestions for Authors

The article by Samrit, T. et al. about the histopathological features after the administration of different oil supplements in rats seems scientifically correct. I think the text has many issues, but most of them can be easily solved.

I have only a major comment to make:

- Personally, I think the discussion is too long and discusses several already-well-known issues. This way, the important facts of the experiment are diluted into meaningless text. I think you can use here the motto: “less is more”. Please, remain sticked to your experiment in the whole discussion.

There are also some minor comments:

- Introduction, page 1, first paragraph 8th line. It seems the word “liver” has little sense before [4] reference.

- Introduction, page 1, first paragraph.  The following sentence (“The source of fatty acids…”) also has little sense, please revise. Something like “fatty acids are produced from…”.

- Introduction, page 2, second paragraph. Please introduce Plukenetia: Sacha inchi oil is extracted from Plukenetia volubilis, that has properties or is characterized by…. “Moreover, Plukenetia…” is not a coherent way of presenting this gender to the reader. After describing the plant and the oil, employ only one term to refer to the oil (SIO or Plukenetia seed oil). All this paragraph should be redacted again. Additionally, you mention Plukenetia huayabambana in the discussion.

- Introduction, page 2, third paragraph. The study was aimed to. Thereafter, when you describe olive oil as EVOO you should include “extra virgin” before olive oil.

- Results, page 4, second paragraph. The results are shown in figure 2.

- Results, page 5, first paragraph. The weight of the rats was treated with an oil supplement? May be you treated the rats, but their weight? In addition, I am confuse with the reference to 8 weeks in the next sentence; according to Figure 3A, the greatest difference in body weight gain with regard to SIO group appears between 8 and 12 weeks…  May be you need to change some words in this sentence to be clearer. Table 2 refers to 8 weeks or 12?

- Results, page 5, second paragraph. Sentence “All oil supplements did not significantly…” needs refinement.

- Results, page 7, first paragraph. You should specify you are referring to the zone 2 and zone 3 of the liver lobule. You can include a histopathology reference to illustrate what you are referring. In the next sentence I miss a word after “fatty”, may be you can include droplets or deposits. The experiment is good, but the redaction needs many improvements… Finally, I miss a figure here to illustrate the fatty change, may be just a high power view of the 4 groups described in section 2.5.

- Results, page 7, second paragraph. TG were measured.

- Results, page 7. You should revise the title of section 2.7.

- Results, page 7, third paragraph. The first sentence is not correct, as Figure 7 correspond to the microscopic examination, you should include (data not shown) or include a macroscopic image of the organs (that can be a supplementary figure); Figure 7 reference should be after the current fourth sentence. By the way, the second sentence in this paragraph has no meaning and is confuse, you can remove it. In addition, please, specify in the text if you detected lipid droplets in the pancreas.

- Results, page 8, Figure 4 legend. Slight fatty accumulation. Images c and d. Images e and f. Images c and d. And so on.

- Results, page 10. I think you can include Figure 7 as a Supplementary figure, but it is a personal opinion, you are not required to do it.

- Discussion, page 11. The first paragraph is extremely long. I can´t understand the third sentence, which in addition should have supporting references. The following sentence about the olives should be in past tense, and you should mention this information as the results of other groups. The following sentence comparing Peru SIO should include the origin of your SIO to be a proper comparison.

- Discussion, page 11, first paragraph (second part). I don´t know why linoleic acid deserves capital letter and linolenic acid does not. Linoleic acid is converted to whatever by a certain enzyme (it can´t be converted by itself). Inflammation  and anti-inflammation factors.

- Discussion, page 11, first paragraph (third part). If omega-6 exceeds omega-3 level sentence needs a reference; two sentences ahead you state “Conversely”, which is wrong, as in this sentence you are stating exactly the same (you can merge both sentences).

- Discussion, page 12, second paragraph. In your experiment, oil supplements actually induced steatosis in the liver (particularly LO), this is actually an influence in the body composition. It may be the only influence, if you want. This paragraph is long, as I mentioned before, the actual discussion is diluted.

- Discussion, pages 12 and 13 paragraph. They can prevent. This paragraph is also extremely long. The part at the end of this page and beginning of the next, describing the liver lobule and zones should be in the introduction section. What´s squalene??? I can see no connection with your experiment. Why do you mention ATP? There is a lot of meaningless text.

- Discussion, page 13, second paragraph. This is an example of a good paragraph. In this case you remain sticked to your experiment and discuss it. Nice.

-  Materials and methods, page 14, first paragraph. The LO (…) was produced in Thailand in 2021 and purchased from a supermarket.

- Materials and methods, page 15, second paragraph. I can´t understand this sentence: “The normal diet in each cage was weight every week”. Rats were necropsied.

- Materials and methods, page 15, last paragraph. I think that if you mention “dehydration”, you don´t need to describe how the method is performed, as is a common procedure regarding histological techniques. You can keep it if you want, but refine the first sentence.

- Materials and methods, page 16, first paragraph. Until the water WAS clear. When you mention Olympus you should include city and country. Morphology and pathological features were determined by… in all samples. 6 fields from each section were randomly selected… is a sentence that also needs to be rewritten. In addition, the individuals who interpreted the pathology of the liver should be mention (I guess they are authors of this work, you can include their initial capital letters).

- Conclusions should be brief, this section is a very brief summary. It is not the place to explain the aim of the study (this is already done before). I could accept the conclusions section as is if you remove the first and the last sentence of the paragraph. However, if you perform even briefer, it would be better. For example, you could include the sentence presented as a summary (the last), but I think you can employ another sentence (two or three sentences for your experiment may be fine).

Comments on the Quality of English Language

The language is not good, with various issues, some of them mentioned as minor faults. At some points I can´t understand what authors want to say. I think the text needs refinement in both the composition and the grammar. A professional revision will help decisively.

Author Response

Dear Professor

Thank you very much for your kind consideration, comments, and suggestions. All the suggested corrections / additions in the revised manuscript have been made and colored blue. Followings are answers to the reviewers’ comments: Comments and Suggestions for Authors

The article by Samrit, T. et al. about the histopathological features after the administration of different oil supplements in rats seems scientifically correct. I think the text has many issues, but most of them can be easily solved.

I have only a major comment to make:

Personally, I think the discussion is too long and discusses several already-well-known issues. This way, the important facts of the experiment are diluted into meaningless text. I think you can use here the motto: “less is more”. Please, remain sticked to your experiment in the whole discussion.

There are also some minor comments:

Comment 1 Introduction, page 1, first paragraph 8th line. It seems the word “liver” has little sense before [4] reference.

Response: We have made the adjustments based on your suggestions (line 39).

Comment 2 Introduction, page 1, first paragraph.  The following sentence (“The source of fatty acids…”) also has little sense, please revise. Something like “fatty acids are produced from…”.

Response: We agree with your suggestions and have already made the revisions. (line 39-40).

Comment 3 Introduction, page 2, second paragraph. Please introduce Plukenetia: Sacha inchi oil is extracted from Plukenetia volubilis, that has properties or is characterized by…. “Moreover, Plukenetia…” is not a coherent way of presenting this gender to the reader. After describing the plant and the oil, employ only one term to refer to the oil (SIO or Plukenetia seed oil). All this paragraph should be redacted again. Additionally, you mention Plukenetia huayabambana in the discussion.

Response: We agree with your excellent suggestions and have enthusiastically implemented the changes (line 75-79).

Comment 4 Introduction, page 2, third paragraph. The study was aimed to. Thereafter, when you describe olive oil as EVOO you should include “extra virgin” before olive oil.

Response: We have implemented the changes as per your recommendations (line 96).

Comment 5 Results, page 4, second paragraph. The results are shown in figure 2.

Response: We have made the text revisions as suggested (line 122).

Comment 6 Results, page 5, first paragraph. The weight of the rats was treated with an oil supplement? May be you treated the rats, but their weight? In addition, I am confuse with the reference to 8 weeks in the next sentence; according to Figure 3A, the greatest difference in body weight gain with regard to SIO group appears between 8 and 12 weeks…  May be you need to change some words in this sentence to be clearer. Table 2 refers to 8 weeks or 12?

Response: We have revised the sentence correctly according to your valuable suggestions (line 149-152).

Comment 7 Results, page 5, second paragraph. Sentence “All oil supplements did not significantly…” needs refinement.

Response: We have successfully made the revisions to the text. (line 155-156).

Comment 8 Results, page 7, first paragraph. You should specify you are referring to the zone 2 and zone 3 of the liver lobule. You can include a histopathology reference to illustrate what you are referring. In the next sentence I miss a word after “fatty”, may be you can include droplets or deposits. The experiment is good, but the redaction needs many improvements… Finally, I miss a figure here to illustrate the fatty change, may be just a high power view of the 4 groups described in section 2.5.

Response: Thank you for the guidance that has helped enhance our work. We have enthusiastically updated the content on liver zonation and added a high-power view image as per your valuable suggestion (line 176-180 and figure 4).

Comment 9 Results, page 7, second paragraph. TG were measured.

Response: We have made the text revisions as suggested (line 210).

Comment 10 Results, page 7. You should revise the title of section 2.7.

Response: We change the title of section 2.7 toSafety of sacha inchi oil on the histology of kidneys, pancreas, and spleen in rats(line 221).

Comment 11 Results, page 7, third paragraph. The first sentence is not correct, as Figure 7 correspond to the microscopic examination, you should include (data not shown) or include a macroscopic image of the organs (that can be a supplementary figure); Figure 7 reference should be after the current fourth sentence. By the way, the second sentence in this paragraph has no meaning and is confuse, you can remove it. In addition, please, specify in the text if you detected lipid droplets in the pancreas.

Response: We have revised the sentence according to your suggestion (line 222-223).

Comment 12 Results, page 8, Figure 4 legend. Slight fatty accumulation. Images c and d. Images e and f. Images c and d. And so on.

Response: We have updated the description of Figure 4 following your suggestions (line 201-208).

Comment 13 Results, page 10. I think you can include Figure 7 as a Supplementary figure, but it is a personal opinion, you are not required to do it.

Response: Thank you for your valuable advice.

Comment 14 Discussion, page 11. The first paragraph is extremely long. I can´t understand the third sentence, which in addition should have supporting references. The following sentence about the olives should be in past tense, and you should mention this information as the results of other groups. The following sentence comparing Peru SIO should include the origin of your SIO to be a proper comparison.

Response: Thank you immensely for your valuable advice. We have made the necessary edits and removed the less critical sections (line 251-257, and 260-269).

Comment 15 Discussion, page 11, first paragraph (second part). I don´t know why linoleic acid deserves capital letter and linolenic acid does not. Linoleic acid is converted to whatever by a certain enzyme (it can´t be converted by itself). Inflammation and anti-inflammation factors.

Response: We have updated the discussion section following your valuable suggestions (line 276-282).

Comment 16 Discussion, page 11, first paragraph (third part). If omega-6 exceeds omega-3 level sentence needs a reference; two sentences ahead you state “Conversely”, which is wrong, as in this sentence you are stating exactly the same (you can merge both sentences).

Response: We have enthusiastically added references and made revisions as per your valuable guidance (line 300-301).

Comment 17 Discussion, page 12, second paragraph. In your experiment, oil supplements actually induced steatosis in the liver (particularly LO), this is actually an influence in the body composition. It may be the only influence, if you want. This paragraph is long, as I mentioned before, the actual discussion is diluted.

Response: We agree with your excellent suggestions and have made the revisions in the discussion section accordingly (line 343-345).

Comment 18 Discussion, pages 12 and 13 paragraph. They can prevent. This paragraph is also extremely long. The part at the end of this page and beginning of the next, describing the liver lobule and zones should be in the introduction section. What´s squalene??? I can see no connection with your experiment. Why do you mention ATP? There is a lot of meaningless text.

Response: We have removed the excessively detailed content as you suggested (line 383).

Comment 19 Discussion, page 13, second paragraph. This is an example of a good paragraph. In this case you remain sticked to your experiment and discuss it. Nice.

Response: Thank you for the valuable advice and for helping to further improve our work.

Comment 20 Materials and methods, page 14, first paragraph. The LO (…) was produced in Thailand in 2021 and purchased from a supermarket.

Response: We have revised the text in LO detail following your suggestion (line 424-425).

Comment 21 Materials and methods, page 15, second paragraph. I can´t understand this sentence: “The normal diet in each cage was weight every week”. Rats were necropsied.

Response: We have implemented the changes as per your recommendations (line 487).

Comment 22 Materials and methods, page 15, last paragraph. I think that if you mention “dehydration”, you don´t need to describe how the method is performed, as is a common procedure regarding histological techniques. You can keep it if you want, but refine the first sentence.

Response: We have removed the unnecessary content as per the suggestion (line 513).

Comment 23 Materials and methods, page 16, first paragraph. Until the water WAS clear. When you mention Olympus you should include city and country. Morphology and pathological features were determined by… in all samples. 6 fields from each section were randomly selected… is a sentence that also needs to be rewritten. In addition, the individuals who interpreted the pathology of the liver should be mention (I guess they are authors of this work, you can include their initial capital letters).

Response: We have added information following your valuable suggestions (line 525-528).

Comment 24 Conclusions should be brief, this section is a very brief summary. It is not the place to explain the aim of the study (this is already done before). I could accept the conclusions section as is if you remove the first and the last sentence of the paragraph. However, if you perform even briefer, it would be better. For example, you could include the sentence presented as a summary (the last), but I think you can employ another sentence (two or three sentences for your experiment may be fine).

Response: We are immensely grateful for your advice. We agree with your suggestions and have already made the revisions (line 565-567).

Sincerely yours,

Tepparit Samrit

Round 2

Reviewer 2 Report

Comments and Suggestions for Authors

The manuscript by Samrit, T. et al. included reviewer´s suggestions and was notably improved. The text is clear now. However, I feel that paragraphs are still too long.  For example, the first paragraph in the discussion is more than a page… just a single paragraph!

There are few minor comments:

- Results, page 3, line 103. You should refer to Figure 1 in the main text (probably at the same time you refer to Table 1).

- Results, page 7, line 176. The histopathology of the liver was assessed though H&E and Oil Red O staining…

- Results, page 7, line 181. As long as the different zones of the hepatic lobule are already defined, you can only mention zone 2 and zone 3. You can remove “around the middle” and “portal vein”. The same in line 185, you can remove the reference to “portal triad” and write just zone 1. I think, it is clearer this way. In line 187, you can include “just” before “had a microvesicular”.

- Discussion, page 11. The first paragraph is still extremely long.

-  Materials and methods, page 14, line 424. The LO was produced in Thailand AND purchased from a supermarket in 2021.

- Materials and methods, page 16, line 528. What´s “based on T.S.”?

Comments on the Quality of English Language

The language was notably improved, but I think there are still some minor faults. The style can be improved separating the long paragraphs into smaller ones.

Author Response

Dear Professor

Thank you very much for your kind consideration, comments, and suggestions. All the suggested corrections / additions in the revised manuscript have been made and colored blue. Followings are answers to the reviewers’ comments:

Comments and Suggestions for Authors

The manuscript by Samrit, T. et al. included reviewer´s suggestions and was notably improved. The text is clear now. However, I feel that paragraphs are still too long.  For example, the first paragraph in the discussion is more than a page… just a single paragraph!

There are few minor comments:

üComment 1 Results, page 3, line 103. You should refer to Figure 1 in the main text (probably at the same time you refer to Table 1).

Response: We have made the adjustments based on your suggestions (line 103).

üComment 2 Results, page 7, line 176. The histopathology of the liver was assessed though H&E and Oil Red O staining.

Response: We agree with your suggestions and have already made the revisions. (line 176).

üComment 3 Results, page 7, line 181. As long as the different zones of the hepatic lobule are already defined, you can only mention zone 2 and zone 3. You can remove “around the middle” and “portal vein”. The same in line 185, you can remove the reference to “portal triad” and write just zone 1. I think, it is clearer this way. In line 187, you can include “just” before “had a microvesicular”.

Response: We agree with your excellent suggestions and have enthusiastically implemented the changes (line 181 and 187).

üComment 4 Discussion, page 11. The first paragraph is still extremely long.

Response: We have implemented the changes based on your recommendations by separating the first paragraph in the discussion section to three paragraphs for easier reading and providing more specific details in each section. (line 275 and 295).

üComment 5 Materials and methods, page 14, line 424. The LO was produced in Thailand AND purchased from a supermarket in 2021.

Response: We have made the text revisions as suggested (line 424-425).

üComment 6 Materials and methods, page 16, line 528. What´s “based on T.S.”?

Response: We have revised the sentence correctly according to your valuable suggestions. We have identified the author who analyzed the results as Tepparit Samrit (line 529).

Comments on the Quality of English Language

üComment 7 The language was notably improved, but I think there are still some minor faults. The style can be improved separating the long paragraphs into smaller ones.

Response: Thank you so much, we have implemented the changes based on your recommendations by separating the first paragraph in the discussion section to three paragraphs for easier reading and providing more specific details in each section. (line 275 and 295).
